# Feruloylation and Hydrolysis of Arabinoxylan Extracted from Wheat Bran: Effect on Dough Rheology and Microstructure

**DOI:** 10.3390/foods13152309

**Published:** 2024-07-23

**Authors:** Solja Pietiäinen, Youngsun Lee, Amparo Jimenez-Quero, Kati Katina, Ndegwa H. Maina, Henrik Hansson, Annelie Moldin, Maud Langton

**Affiliations:** 1Department of Molecular Sciences, Swedish University of Agricultural Sciences, Almas Allé 5, 750 07 Uppsala, Sweden; kati.katina@helsinki.fi (K.K.); henrik.hansson@slu.se (H.H.); maud.langton@slu.se (M.L.); 2Lantmännen ek för, Sankt Göransgatan 160, 112 17 Stockholm, Sweden; annelie.moldin@lantmannen.com; 3Department of Food and Nutrition, University of Helsinki, Agnes Sjöbergin katu 2, 00014 Helsinki, Finland; youngsun.lee@helsinki.fi (Y.L.); henry.maina@helsinki.fi (N.H.M.); 4Division of Industrial Biotechnology, Department of Life Sciences, Chalmers University of Technology, 412 96 Gothenburg, Sweden; amparo@chalmers.se; 5Division of Glycoscience, Department of Chemistry, KTH Royal Institute of Technology, AlbaNova University Centre, SE-106 91 Stockholm, Sweden

**Keywords:** rheology, arabinoxylan, hydrolysis, ferulic acid, dough

## Abstract

Feruloylated arabinoxylan (AX) is a potential health-promoting fiber ingredient that can enhance nutritional properties of bread but is also known to affect dough rheology. To determine the role of feruloylation and hydrolysis of wheat bran AX on dough quality and microstructure, hydrolyzed and unhydrolyzed AX fractions with low and high ferulic acid content were produced, and their chemical composition and properties were evaluated. These fractions were then incorporated into wheat dough, and farinograph measurements, large and small deformation measurements and dough microstructure were assessed. AX was found to greatly affect both fraction properties and dough quality, and this effect was modulated by hydrolysis of AX. These results demonstrated how especially unhydrolyzed fiber fractions produced stiff doughs with poor extensibility due to weak gluten network, while hydrolyzed fractions maintained a dough quality closer to control. This suggests that hydrolysis can further improve the baking properties of feruloylated wheat bran AX. However, no clear effects from AX feruloylation on dough properties or microstructure could be detected. Based on this study, feruloylation does not appear to affect dough rheology or microstructure, and feruloylated wheat bran arabinoxylan can be used as a bakery ingredient to potentially enhance the nutritional quality of bread.

## 1. Introduction

Wheat bran is a low-value by-product form wheat processing that has an estimated annual global production volume of 150 million tons [1]. It is currently utilized mostly as animal feed even though it contains many health-promoting components, such as dietary fiber, that could be used as food ingredients to increase the value of wheat bran side stream [2]. Arabinoxylan (AX) is an abundant dietary fiber component in wheat bran composed of a β-(1→4)-linked β-D-xylopyranose backbone substituted with an arabinofuranosyl group [3]. Arabinose can be further linked to ferulic acid (FA) via an ester bond [4], and feruloylated AX has been shown to have antioxidant properties even after baking and fermentation [5,6]. This suggests that feruloylated AX could offer further health benefits when added to bread, as food antioxidants and their anti-inflammatory properties have been linked to the prevention of cardiovascular diseases and cancer [7]. Despite the availability of wheat bran and the well-known health benefits of increased dietary fiber intake, wheat bran AX is currently not utilized as a bread ingredient partly due to its negative effect on bread quality [8]. Like many fibers, also AX can have a detrimental effect on bread quality, often leading to decreased volume and denser bread crumb especially with fiber addition levels of 5% or above [8]. New approaches are therefore required to enable the use of feruloylated wheat bran AX as a bread ingredient.

We have recently demonstrated how hydrolysis and feruloylation of wheat bran AX can improve its properties as bread ingredient, as feruloylated and hydrolyzed arabinoxylan produced the bread closest to control in volume and crumb structure [9]. This indicates that modifying AX structure and properties in terms of feruloylation and hydrolysis can help improve the quality of bread with incorporated fiber. However, the effect feruloylation and hydrolysis of AX on bread quality remains unclear as contrary results exist. The mechanical properties of the dough govern the quality of bread [10] and understanding of dough quality can therefore provide valuable insights into the bread-baking quality of feruloylated AX. Some studies suggest that FA can strengthen gluten network via covalent cross-linking [11,12,13,14] when others have observed addition of FA to reduce the amount of disulfide bonds and glutenin macropolymer content [15] and increase dough firmness with lower concentrations [16]. Wang et al. [17] observed an improvement in dough extensibility with external ferulic acid, suggesting that wheat bran AX with low ferulic acid content directly decreases the extensibility of dough due to less cross-linking between gluten and AX. However, many of the previously mentioned studies used free FA rather than feruloylated AX. Previous research has demonstrated that reducing molar mass through hydrolysis is crucial for AX functionality in dough [18,19], but to our knowledge its connection to feruloylation and their combined effect on AX dough rheology and microstructure has not been studied previously. More knowledge on the connection between AX feruloylation, hydrolysis, and dough properties are therefore needed to understand the effect of AX structure on dough quality and hence facilitate use of feruloylated AX as a bread ingredient.

The aim of this study was to determine the effect of partial hydrolysis and feruloylation of wheat bran AX on wheat dough rheology and microstructure. Unhydrolyzed and hydrolyzed AX with high and low levels of FA were incorporated in dough. Dough water absorption, development time and stability were determined using a farinograph and dough extensibility and rheological characteristics were measured in large and small deformations. Microstructure of doughs was observed using light microscopy. This work provides new insights into how feruloylation and hydrolysis of AX affect dough properties and microstructure and therefore enable the use of wheat bran AX as a functional bread ingredient to enhance the technological and nutritional quality of bread with incorporated fiber.

## 2. Materials and Methods

### 2.1. Materials

Feruloylated wheat bran arabinoxylan (FAX) was supplied by Lantmännen (Stockholm, Sweden). The process to produce this fraction in pilot scale has been previously detailed by Zhang et al. [5]. Wheat flour (Special Vetemjöl, Lantmännen, Sweden) was bought from local supermarket. All chemicals and reagents were purchased from Sigma-Aldrich (St. Luis, MO, USA) and were analytical grade, unless otherwise stated.

### 2.2. Preparation of Arabinoxylan Fractions

#### 2.2.1. Removal of Ferulic Acid

To produce a fraction without ferulic acid (AX), original FAX fraction was subjected to mild saponification using a process from Zhang et al. [5] with modification. FAX was mixed with 0.5 M NaOH and sample was stirred 4 h 20 °C. After saponification, pH was adjusted to pH 7 with 0.5 M acetic acid. AX was precipitated with ethanol (99.6%, 1:4 *v*/*v*) by stirring for 2 h followed by cooling overnight (4 °C). Arabinoxylan was filtered through a porous metal plate (40 μm) washed 3 times with 75% (*v*/*v*) ethanol and freeze dried.

#### 2.2.2. Hydrolysis of Arabinoxylan

To produce hydrolyzed fractions (H-FAX and H-AX), FAX and AX were enzymatically hydrolyzed using a process previously described and optimized by Ruthes et al. [20]. First the fraction were first solubilized in water by using a solid:liquid ratio of 1:10, adjusting pH to pH 5 using 0.5 M acetic acid and then heating to 60 °C under constant mixing. Then Pentopan Mono BG, a 1,4-β-xylanase (2500 FXU-W/g, Novozymes, Lyngby, Denmark) was added at 20 U/g AX. After incubation for 24 h in 37 °C, the enzyme was inactivated by heating to 100 °C for 5 min and fractions were then freeze dried.

### 2.3. Characterization of Fraction Composition

#### 2.3.1. Monosaccharide Composition and Klason Lignin Content

The monosaccharide composition was determined from AX fractions as previously described by Lu et al. [21] with modifications. 1 mL 72% H_2_SO_4_ was added to 70 mg of sample and kept under vacuum for 1.5 h in 20 °C. After incubation, 29 mL of water was added, and the samples were autoclaved at 125 °C for 1 h. The samples were then vacuum filtered with 10 mL of hot water and diluted. For reference, a standard solution containing glucose, arabinose, xylose, rhamnose and mannose was prepared the same way as samples. Monosaccharide composition was analyzed using HPAEC with a pulsed amperometry detector (ICS 3000 Dionex, Thermo Scientific, Sunnyvale, CA, USA) equipped with an AEC column (CarboPac PA1 guard 4 × 50 mm and CarboPac PA 1 analytical 4 × 250 mm, Thermo Scientific, Sunnyvale, CA, USA). Klason lignin content was determined as the acid-insoluble residue after hydrolysis.

#### 2.3.2. Basic Nutrient Composition

The protein content of fractions was measured by their total N content using the Dumas combustion method in triplicate with a factor of 6.25 applied to calculate the protein content. Ash content was assessed in triplicate following the AACC total ash method [22] The total starch content of the AX fractions was determined using a resistant starch assay kit (Resistant Starch Assay Kit (Rapid), Megazyme Ltd., Wicklow, Ireland), with the total starch content calculated as the sum of resistant and digestible starch. The β-glucan content was measured in triplicate with a β-glucan assay kit (Mixed Linkage Assay Kit, Megazyme Ltd., Wicklow, Ireland).

#### 2.3.3. Ferulic Acid Content

The ethanol-soluble free and conjugated ferulic acid, as well as the ethanol-insoluble esterified bound ferulic acid content of fractions, were extracted and quantified as previously described by Li et al. [23] with modifications. All samples were prepared in triplicates. 100 mg of sample was extracted with 80% ethanol, sonicated for 10 min, and the supernatant was collected after centrifugation. This process was repeated three times, and the combined supernatants were evaporated under nitrogen (extract A). For conjugated ferulic acid, dried extract A was hydrolyzed with 800 µL 2 M NaOH (Merk KGaA, Darmstadt, Germany) and incubated for 16 h after oxygen removal. The pH was then adjusted to pH 2 using 12 M HCl, and the conjugated ferulic acids were extracted with ethyl acetate three times (extract B). For the bound ferulic acid, the residual pellet obtained after extraction with 80% ethanol, was hydrolyzed with 2 M NaOH (Merk KGaA, Darmstadt, Germany) followed by 16 h incubation at 20 °C. After centrifugation, the collected supernatant was adjusted to pH 2 with 12 M HCl followed by the addition of approximately 30 mg of NaCl (≥99.5%, Merk KGaA, Darmstadt, Germany), and then extracted with ethyl acetate three times (extract C). All ferulic acid extracts were evaporated with nitrogen, re-dissolved in 10% methanol and centrifuged (14,000 × *g*, 15 min, 20 °C) using Amicon filter (0.5 mL, 10 K, Merck Millipore Ltd., Co., Cork, Ireland) before analysis. Extract A was used as is for free ferulic acid, extract B for conjugated ferulic acid and extract C for bound ferulic acid.

The ferulic acid standard was purchased from Sigma-Aldrich (Merck KGaA, Darmstadt, Germany) and the stock solution was prepared in methanol (1 mg/mL). The stock solution was diluted with Milli-Q water and utilized for preparation of the calibration solution. Ferulic acid content was quantified using an ACQUITY ultra high-performance liquid chromatography system with a photodiode array detector (UPLC-PDA, Waters, Milford, MA, USA) at a wavelength of 320 nm. Separation was conducted using an ACQUITY UPLC HSS T3 column (1.8 µm, 2.1 × 150 mm, Waters) connected with a VanGuaurd precolumn (2.1 × 5 mm, Waters), at 40 °C. Mobile phases A and B were 0.5% formic acid in Milli-Q water and Acetonitrile, respectively. The flow rate was 0.5 mL/min, and the linear gradient was as follows: 0 min, 90% A; 10 min, 80% A; 14 min, 10% A, 15 min, 10% A; 16 min, 90% A with a re-equilibrum time of 4 min. The data was processed by Empower 3 (Waters) and Excel (Microsoft).

### 2.4. Characterization of Fraction Properties

#### 2.4.1. Molar Mass Distribution

Molar mass distribution was analyzed using size exclusion chromatography (SEC) (SECurity 1260, Polymer Standard Services, Mainz, Germany), following the method from Ruthes et al. [20].

#### 2.4.2. Water Holding Capacity of Fractions

Water holding capacity (WHC) of fractions was determined in triplicate according to method described previously by Hemdane et al. [24] with modifications. 0.5 g of sample was weighted into a tube with 5 mL of water, and left to room temperature for 60 min after mixing. Samples were then centrifuged (10,000× *g* 10 min at 20 °C). The supernatants were separated and pellets were turned upside down for 15 min to remove excess water. WHC was expressed as g of water retained by 1 g of dry matter.

#### 2.4.3. Rheological Properties of Fractions

Rheological properties of fractions were determined from AX solutions that were prepared by mixing fractions with water (4% *w*/*v*) and stirring continuously at 60 °C for 2 h. Rheological properties were measured using a Discovery HR-3 rheometer (TA Instruments, New Castel, DE, USA) with a 40 mm cone plate. Flow curves were obtained at a shear rate range from 0.01 to 1000 1/s.

### 2.5. Dough Quality

#### 2.5.1. Water Absorption, Dough Development Time (DDT) and Dough Stability

Water absorption, dough development time (DDT) and dough stability were measured for wheat flour and mixtures of flour and 5% of arabinoxylan fractions with a farinograph (Brabender GmhH, Duisburg, Germany) following AACC method 54-21.01 [25]. 5% addition level was chosen based on our previous study on the effect of feruloylated and hydrolyzed AX fractions on bread quality [9], as lower addition levels did not result in clear differences in bread quality. Fractions were added by replacing flour with arabinoxylan based on fractions’ AX content. The fractions were premixed with water and heated to 80 °C under constant stirring, and then cooling to room temperature before dough preparation. Water absorption was defined as the amount of water required to reach 500 BU and expressed as percentage of flour weight. DDT was defined as the time from water addition to the dough reaching peak consistency.

#### 2.5.2. Large Deformation Rheological Measurements

Uniaxial extension measurements at large deformations were performed using a texture analyzer (TA-XT Plus, Stable Micro System, Surrey, UK) equipped with a Kieffer dough/gluten extensibility rig (Stable Micro System, Surrey, UK). Control dough was prepared using only flour and water. Doughs containing AX fractions were prepared by adding 5% of fraction by replacing flour based on fractions’ AX content. The fractions were dispersed in water prior to dough preparation by stirring while heating to 80 °C and cooling to room temperature. Doughs were then mixed in farinograph using optimal water absorption and DDT obtained from farinograph results. After mixing in farinograph, doughs were gently rolled into balls and placed relaxing in incubator 30 °C for 20 min. After relaxing, a piece of dough was cut from the center of the dough ball and rolled gently into a sausage shape and placed into an oiled molder and compressed into dough strips. The molder containing the dough strips was placed back into incubator for 40 min 30 °C. After relaxing, a dough strip was removed from the molder and clamped between the plates of the Kieffer rig prior to each test. The samples were tested using 2.0 mm/s test speed and 75.0 mm distance. 6 dough strips were tested from each dough.

#### 2.5.3. Small Deformation Rheological Measurements

Small deformation rheological measurements were performed by oscillation test using a Haake MARS 40 rheometer (ThermoScientific, Sunnyvale, CA, USA) with parallel-plate geometry (35 mm). The doughs were prepared the same way as for dough extensibility test (Section 2.5.2). Dough piece was placed between the plates and pressed to 1 mm gap. Then excess dough was removed and 200/50 cS fluid (Dow Corning Corporate, Midland, MI, USA) was applied to sample edges to prevent sample drying. After 2 min resting time, an oscillation test was done using 0.01% strain based on the linear viscoelastic region of the samples at 25 °C in the frequency range of 0.05–50 Hz. The rheological characteristics were expressed as the storage (G′) and loss modulus (G″).

#### 2.5.4. Dough Microstructure

Dough samples (5 × 5 × 5 mm) were cut from doughs prepared for small deformation rheological tests (Section 2.5.3) and fixated in glutaldrehyde (2.5%) for 24 h. Samples were then dehydrated with a series of ethanol in increasing concentrations and infiltrated and hardened using Technovit 7100 (KULZER, Hanau, Germany). Hardened samples were sectioned into 5 μm sections with a ultramicrotome (Leica Microsystems GmbH, Leica EM UC6, Wetzlar, Germany), stained with light green and examined with a microscope (Nikon, Ecplise Ni-U, Tokio, Japan) equipped with a 40× (0.75 NA) apochromatic objective. Images were captured with Nikon Digital Sight DS-Fi2 camera (Nikon, Japan).

### 2.6. Experimental Design and Statistical Analysis of Data

All measurements were conducted at least in duplicate, with results reported as mean values ± standard deviation. For dough and fraction properties, type-III ANOVA was used to identify differences at a 95% confidence level, determined by Tukey’s pairwise comparisons. The relationship between fraction and dough properties was examined using 2-tailed Pearson correlation with linear relationships evaluated through regression analysis. All data analyses were performed using R (version R 4.3.0, The R Foundation for Statistical Computing, Vienna, Austria), unless otherwise stated.

## 3. Results & Discussion

### 3.1. Fraction Composition

Chemical composition of fractions is presented in Table 1. The fraction modification with hydrolysis or saponification did not affect relative AX content (*p* > 0.05), and AX content was 70.7–73.0% of total carbohydrates for all fractions. However, the amount of total carbohydrates did vary between fractions, with FAX having the highest AX content of 75.9%, followed by AX, H-AX and H-FAX with AX contents of 69.4, 67.5 and 56.2%, respectively. H-FAX had a relatively low amount of carbohydrates compared to other fractions. The results of monosaccharide composition are known to be closely related to the hydrolysis process [26], indicating that sample material might have affected the acid hydrolysis of H-FAX. This difference in AX content should be taken into account when considering results for dough quality. The fractions were added to doughs based on their AX content, so doughs with H-FAX also contained higher levels of other fraction components compared to other fractions. The A/X ratio decreased from 0.36 for FAX to 0.26–0.27 for H-FAX, AX and H-AX, indicating that processing of fractions decreased the amount of arabinose substitution. The ash content was 4.7 and 4.4 for FAX and H-FAX, and 7.1 and 7.3 for AX and H-AX, respectively, indicating that ash content was increased by alkali treatment, as reported previously by Rasool et al. [27]. Glucose, starch, protein and Klason lignin contents were similar between fractions.

FA content of fractions is presented in Figure 1. FAX had the highest amount of FA, 11.2 mg/g, of which almost all were bound FA. Hydrolysis reduced the amount of FA slightly to 8.8 mg/g for H-FAX. For H-FAX, 44% of FA was conjugated, indicating that hydrolysis has converted a portion of the insoluble FA bound to AX into ethanol-soluble conjugated FA. This is likely due to enzyme activity, which hydrolyzes xylose backbone into smaller AX fragments with attached FA. This process can increase ethanol solubility of FA, and therefore increases the proportion of conjugated FA. Fractions with low FA content still contained traces of FA, which indicates that pilot scale saponification was not able to remove FA as efficiently as previously [9]. However, the FA extraction was more detailed and contained 3 extraction steps instead of one, which might also affect the amount of FA detected in samples. Saponification reduced the total FA content to 2.0 and 1.5 mg/g for AX and H-AX, respectively. Like H-FAX, also H-AX contained less bound and more conjugated FA compared to unhydrolyzed AX. In this study, the focus was only on the differences in total amount of FA and the differences in the type of FA were not considered. However, bound and free FA have been shown to differ in their antioxidant capacity [15], indicating that there could even be differences in the functionality between conjugated and bound FA in breadmaking.

### 3.2. Fraction Properties

Fraction properties in terms of molar mass, WHC and solution viscosity are presented in Table 2. Molar mass was expectedly reduced by hydrolysis, and weight average molar mass reduced from 485 to 67 for FAX and 464 to 95 kg mol^−1^ for AX. Molar mass correlated strongly with both viscosity and WHC of fractions (Appendix A), indicating that hydrolysis decreases both solution viscosity and WHC. The correlation between molar mass and both WHC and viscosity has been widely reported [12,28,29], and the strong tendency of high molar mass AX to absorb water has been shown to play a key role in its functionality in dough. Buksa et al. [29] reported that relative viscosities were positively correlated with the molar mass of arabinoxylan fractions.

### 3.3. Dough Properties

#### 3.3.1. Water Absorption, Dough Development Time (DDT) and Dough Stability

Farinograph results for control dough and doughs prepared with 5% fiber fractions are presented in Table 3. Fiber addition increased water absorption and DDT, and decreased dough stability with all fractions (*p* < 0.05). Hydrolysis decreased water absorption of FAX and AX from 78.5 and 83.3 to 64.5 and 63.3%, respectively (*p* < 0.05). Similar to WHC, also farinograph water absorption correlated strongly (*p* < 0.001) with molar mass and viscosity of fractions (Appendix A). This effect has been shown previosuly by Biliaderis et al. [19].

DDT of FAX and AX was also decreased from 7.0 and 7.8 to 4.9 and 6.2, respectively, and DDT was also found to correlate with molar mass, viscosity and WHC of fraction (Appendix A). Notably, H-FAX had the DDT closest to control despite the high content of other fraction components in H-FAX doughs due to lower AX content in fraction, as discussed in Section 3.1. DDT represent the time required to develop a dough with optimal consistency, and the increased DDT has been linked to the higher water absorption, leading to competition for water between arabinoxylan, starch and gluten, and therefore delaying gluten network formation [30]. H-AX was the only fraction that did not reduce dough stability time (DST) (*p* > 0.05).

#### 3.3.2. Large Deformation Rheological Measurements

Large deformation extensional measurements, while limited in providing fundamental rheological information, are crucial for assessing the strength of wheat flour due to the large deformations encountered during dough processing, such as mixing, sheeting, and baking [31]. Extension curves for control dough and doughs prepared with 5% fiber fractions are presented in Figure 2, where maximum force (N) represents dough resistance to extension and distance to break (mm) represents doughs extensibility. Fiber addition decreased dough extensibility and increased resistance to extension compared to control, indicating that fiber addition increases dough firmness and produces weaker doughs. Hydrolyzed fractions H-FAX and H-AX showed higher extensibility compared to their unhydrolyzed counterparts FAX and AX. As dough extensibility is usually related with dough strength [17], these results show that fiber addition weakened the doughs, and this effect was more severe for unhydrolyzed fractions. Fiber incorporation has been previously shown to decrease extensibility of dough due to breakage of the starch-gluten matrix, restricting the retention of gas in the gluten network and preventing gluten agglomeration [20,32]. As extensibility was also found to correlate negatively with molar mass, viscosity and WHC of fractions (Appendix A), these results further support the existing evidence that the higher WHC, water absorption and viscosity of unhydrolyzed fractions disrupt the gluten network to a larger extent compared to their hydrolyzed counterparts.

H-AX showed the highest resistance to extension but was still able to maintain a similar extensibility compared to control while increasing resistance to extension, doughs prepared with H-AX being firm but relatively strong compared to other fractions. These results were in line with the farinograph results (Section 3.3.1), where H-AX improved the dough stability compared to other fractions. The other hydrolyzed fraction H-FAX was closest to control in terms of resistance to extension and extensibility. This was in line with results from previously published baking trails using similar fractions [9], where H-FAX produced a bread comparable to control at the 5% addition level. This indicates that hydrolyzed fractions disrupt the starch-gluten matrix less compared to their unhydrolyzed counterparts. Hydrolyzed AX fractions might also promote the connectivity of the gluten network by AX-starch-gluten interactions as suggested previously by Li et al. [33]. Even though Wang et al. [17] observed wheat bran AX with low ferulic acid content directly, no difference between feruloylated and unferuloylated fractions was observed in this study.

#### 3.3.3. Small Deformation Rheological Measurements

An oscillation test was performed to provide more fundamental information about the effect of arabinoxylan incorporation on the viscoelastic properties of wheat dough. Figure 3 shows the frequency sweep (G′ and G″) curves of control dough and doughs enriched with AX fractions. Dough is a complex system showing both elastic and adhesive behavior [17]. We found the storage modulus (G′) for all doughs tested to be higher than the loss modulus (G″), showing a predominant elastic behavior. Addition of unhydrolyzed fractions increased G′ compared to control, while hydrolyzed fractions H-FAX and H-AX decreased G′ compared to control. G′ was also correlated with molar mass, viscosity and WHC of fractions (Appendix A). G′ describes the materials ability to store deformation energy in an elastic manner, with higher G′ indicating higher mechanical rigidity. These results indicate that unhydrolyzed AX increases the dough stiffness, while hydrolyzed AX increases dough softness. The addition of AX and bran particles in general has been previously observed to increase the storage modulus of dough [33], which has been linked the high WHC of AX leading to water immobilization during dough resting and hence a stiffer dough [24]. The loss modulus (G″) is associated with dough flow properties, such as extensibility and adhesiveness [17], and higher extensibility is linked to increased dough strength. For G″, sample doughs followed a similar pattern compared to G′, unhydrolyzed fractions having the highest values and hydrolyzed fractions the lowest, but the loss modulus was not correlated to fraction properties.

Small deformation measurements did not directly correlate with the large deformation measurements. This was somewhat expected as in small deformation measurements within the frequency range used in this study (0.05–50 Hz), it has been suggested that protein-protein interactions crucial for dough properties are partly masked by starch-starch and starch-protein interaction [31]. It is therefore likely that the large deformation measurements represent more the protein-protein interactions and the small deformation measurements starch-starch and starch-protein interaction. No clear effect from AX feruloylation on small-scale deformation could be detected between fractions. Even though some authors have seen FA to increase dough softness and therefore decrease the negative effect of fiber addition on dough, Snelders et al. [16] observed that a FA content of 0.1–1.7% was not high enough to see any positive effects from addition of AX oligosaccharides. Therefore, minimal effect could be expected with the low FA amounts in the AX fractions used in this study.

#### 3.3.4. Dough Microstructure

To visualize changes in dough as an effect of fiber incorporation, doughs were visualized using light microscopy, and gluten was stained green to confirm the expected disturbance of gluten network. Captured images from dough samples are presented in Figure 4. For the control dough with no added fiber, the gluten network occupied almost all areas between starch granules and left almost no unstained background area. For unhydrolyzed fractions FAX and AX, the gluten network seemed more disrupted compared to control, and large portion of areas between starch granules were left unstained. Also, longer non-protein and non-starch particles, that were expected to be residual bran particles from AX fractions, were visible in FAX and AX doughs. The presence of these particles might indicate that residual bran particles in unhydrolyzed AX fractions might also cause physical hindrance to dough formation, as previously reported by Molina et al. [34]. In doughs with hydrolyzed fractions H-FAX and H-AX, the amount of unstained background was less than for hydrolyzed fractions, supporting the explanation that hydrolyzed fractions have less effect on the gluten network formation than unhydrolyzed fractions. Previously Zhu et al. [30] have reported AX incorporation in dough to increase the amount of unstained background in light microscopy. This is supported by Frederix et al. [32], who have shown that supernatant viscosity of AX enriched batters affects gluten agglomeration and the impact was more severe for high molar mass AX than low molar mass AX. Döring et al. [35] observed the incorporation of 5% AX in dough to inhibit protein network formation and cause protein agglomeration. They suggested that increased water content leads to dilution of dough and therefore proteins with less stretching ability, also previosuly described by Jekle and Becker [36].

## 4. Conclusions

Inclusion of wheat bran AX was observed to greatly affect the rheology of wheat dough. This effect appeared to be modulated by hydrolysis of AX, which, in turn, correlated with changes in fraction properties. Hydrolysis of AX fractions decreased the molar mass, solution viscosity and water holding capacity of the AX fractions. Additionally, it resulted in reduced water absorption and dough development time, while showing higher extensibility compared to their unhydrolyzed counterparts FAX and AX. Based on large deformation measurements, fiber addition decreased dough extensibility and increased resistance to extension compared to control, with a more profound effect observed for unhydrolyzed fractions. Addition of unhydrolyzed fractions increased and hydrolyzed fractions decreased G′ compared to control, suggesting that unhydrolyzed AX increases the dough stiffness on small deformations. Results from both large and small deformation measurements demonstrate how especially unhydrolyzed fiber fractions produced firm and stiff doughs with poor extensibility, attributed to fiber disturbing formation of gluten network. Notably, fraction properties such as molar mass, viscosity and water holding capacity strongly correlated with dough extensibility and storage modulus, indicating their role in increasing viscosity and water holding by high molar mass AX. Light microscopy indicated that hydrolyzed fractions might disturb the gluten network less compared to unhydrolyzed fractions FAX and AX. While H-FAX produced doughs closest to control in terms of water absorption, dough development time and extensibility, no clear effects from AX feruloylation on dough properties or microstructure could be detected. Based on this study, feruloylation does not affect dough quality and feruloylated wheat bran arabinoxylan can be utilized as a bakery ingredient to potentially improve the quality of bread with incorporated fiber. However, differences in fraction composition make drawing firm conclusions difficult and further work is needed to evaluate the importance of other fraction compounds to the observed results.

## Figures and Tables

**Figure 1 foods-13-02309-f001:**
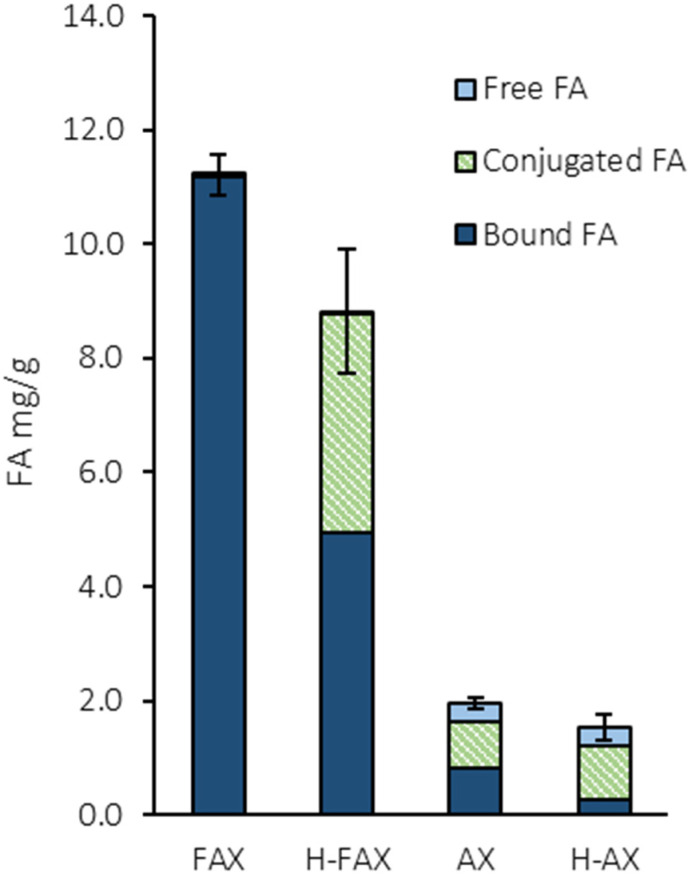
Ferulic acid content of fractions (dwb). Different letters on each bar indicate significant difference (*p* < 0.05). F = feruloylated; H = hydrolyzed.

**Figure 2 foods-13-02309-f002:**
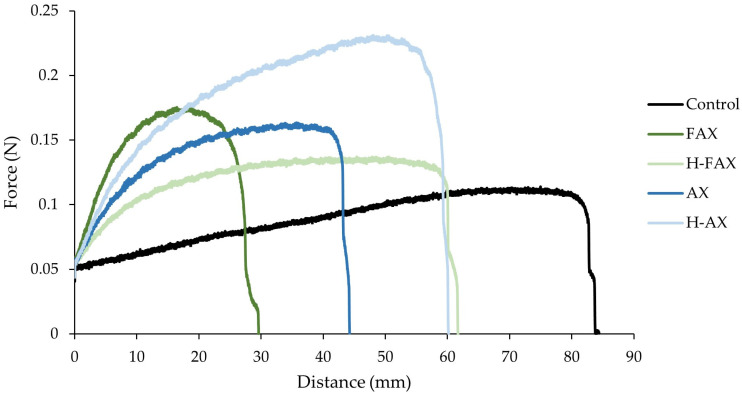
Extension curves with measured force (N) as a function of distance (mm) for control dough and doughs prepared with 5% fiber fractions. F = feruloylated; H = hydrolyzed.

**Figure 3 foods-13-02309-f003:**
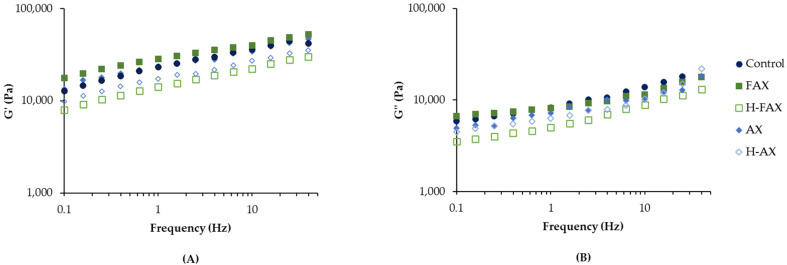
Dough rheological properties from frequency sweep test. Storage modulus G′ (**A**) and loss modulus G″ (**B**) as a function of frequency at 25 °C for doughs with and without fiber incorporation.

**Figure 4 foods-13-02309-f004:**
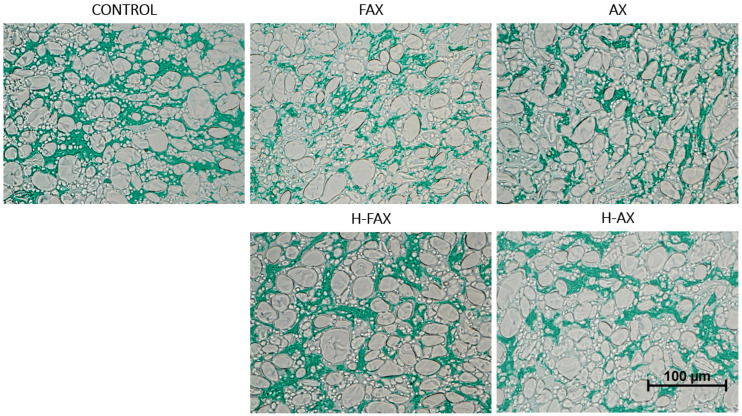
Dough microstructure for control dough and doughs with fiber fractions. Protein stained light green. Magnification 40×. Pixels: 975 × 731.

**Table 1 foods-13-02309-t001:** Carbohydrate, arabinoxylan (AX), starch, mixed linkage β-glucan, Klason lignin, protein, and ash content of the fractions (dwb). Mean value ± SD. F = feruloylated; H = hydrolyzed.

	FAX	H-FAX	AX	H-AX
Total carbohydrates (g/100 g) ^1^	75.9 ± 6.6 ^a^	56.2 ± 13.3 ^b^	69.4 ± 2.6 ^b^	67.5 ± 0.8 ^b^
AX % ^2^	70.7 ± 2.1 ^a^	71.1 ± 0.0 ^a^	73.0 ± 0.2 ^a^	72.0 ± 0.1 ^a^
Glc % ^3^	23.1 ± 2.8 ^a^	26.4 ± 0.1 ^a^	24.5 ± 0.2 ^a^	25.6 ± 0.1 ^a^
Gal % ^3^	3.0 ± 0.4 ^a^	2.5 ± 0.2 ^a^	2.5 ± 0.0 ^a^	2.5 ± 0.0 ^a^
A/X ^4^	0.4 ± 0.1 ^a^	0.3 ± 0.0 ^a^	0.3 ± 0.0 ^a^	0.3 ± 0.0 ^a^
Mixed linkage β-glucan (g/100 g)	5.6 ± 0.6 ^a^	5.3 ± 0.2 ^a^	6.0 ± 1.3 ^a^	5.9 ± 0.1 ^a^
Starch (g/100 g)	0.8 ± 0.0 ^a^	1.0 ± 0.0 ^a^	0.7 ± 0.0 ^a^	1.1 ± 0.1 ^a^
Klason lignin (g/100 g)	3.3 ± 1.0 ^a^	2.6 ± 1.7 ^a^	2.0 ± 1.0 ^a^	3.4 ± 1.1 ^a^
Protein (g/100 g)	2.7 ± 0.0 ^a^	2.6 ± 0.0 ^a^	1.7 ± 0.0 ^b^	1.7 ± 0.0 ^b^
Ash (g/100 g)	4.7 ± 0.0 ^a^	4.4 ± 0.0 ^a^	7.1 ± 0.0 ^b^	7.2 ± 0.0 ^b^

^1^ Total carbohydrate content was calculated based on total content of arabinose, rhamnose, galactose, glucose, xylose, and mannose. ^2^ AX content was calculated based on the % of arabinose and xylose of total carbohydrate content. ^3^ % of total carbohydrate content. Glc = glucose, Gal = galactose. ^4^ Ratio between arabinose and xylose. Different letters indicate significant differences (*p* < 0.05).

**Table 2 foods-13-02309-t002:** Fraction properties in terms of molar mass (number-average molecular weight (Mn), weight-average molecular weight (Mw), and dispersity index (Đ), solution viscosity (mPa.s) and water holding capacity (WHC). Different letters indicate significant difference (*p* < 0.05). F = feruloylated; H = hydrolyzed.

	FAX	H-FAX	AX	H-AX
Mw (kg mol^−1^)	485	67	464	95
Mn (kg mol^−1^)	126	37	218	17
Đ	3.8	2.6	2.1	3.9
Viscosity (mPa.s) ^1^	100 ± 18 ^a^	39 ± 1.0 ^b^	149 ± 44 ^a^	30 ± 8.9 ^b^
WHC ^2^	3.1 ± 0.1 ^a^	0.7 ± 0.2 ^b^	3.3 ± 0.2 ^a^	0.9 ± 0.1 ^b^

^1^ At 1/s. ^2^ g H_2_O/g dry sample.

**Table 3 foods-13-02309-t003:** Water absorption (% of flour weight), dough development time (DDT, min) and dough stability (min) of sample doughs. Mean value ± SD. F = feruloylated; H = hydrolyzed.

	Control	FAX	H-FAX	AX	H-AX
Water absorption (%)	61.0 ± 0.7	78.5 ± 0.6 ***	64.5 ± 0.8 *	83.3 ± 1.3 ***	63.3 ± 0.9
Development time (min)	2.8 ± 0.8	7.0 ± 1.4 *	4.9 ± 0.2	7.8 ± 1.0 *	6.2 ± 0.2
Dough stability (min)	5.2 ± 0.2	2.6 ± 0.1 **	2.1 ± 0.2 **	2.4 ± 0.6 **	4.1 ± 0.4

*p*-values below 0.05 (*), 0.01 (**) and 0.001 (***) indicate statistically significant differences at the 95%, 99% and 99.9% conficende level compared to control, respectively.

## Data Availability

The original contributions presented in the study are included in the article, further inquiries can be directed to the corresponding author.

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
