# Peer review of "Feruloylation and Hydrolysis of Arabinoxylan Extracted from Wheat Bran: Effect on Dough Rheology and Microstructure"

_foods, 2024, doi:10.3390/foods13152309_

Round 1

Reviewer 1 Report

Comments and Suggestions for Authors

The manuscript “Feruloylation and hydrolysis of arabinoxylan extracted from wheat bran: Effect on dough quality and microstructure” is interesting to study a dietary fiber and it could affect dough quality. However, the results did not design properly to investigate the details. Based on comparing research, optimization type experiments should be designed. For example.

1.     why adding the samples as 5%? How about 1%-10%?

2.     Hydrolysis time: 24 h, how about 2-36 h?

3.     There is no significantly different between the microstructures of wheat dough.

4.     If AX can be used in wheat dough, what are the usage limits?

5.     The results cannot support the conclusions.

Author Response

Reviewer 1

The manuscript “Feruloylation and hydrolysis of arabinoxylan extracted from wheat bran: Effect on dough quality and microstructure” is interesting to study a dietary fiber and it could affect dough quality. However, the results did not design properly to investigate the details. Based on comparing research, optimization type experiments should be designed. For example.

We would like to thank reviewer #1 for their thoughtful review. We appreciate the detailed comments regarding the experimental design and content of the manuscript that have helped to improve the quality of this work. The optimization experiments both for fiber addition level and hydrolysis process have been previously done by the authors in following publications: Pietiäinen et al. 2024 https://doi.org/10.1016/j.jcs.2024.103920 and Ruthes et al. 2017 https://doi.org/10.1039/c6gc03473j. We have clarified the connection to these studies in the manuscript, see details below in respective comments.

Comment 1:     why adding the samples as 5%? How about 1%-10%?

Response 1: The fiber addition level of 5% was chosen based on our previously published study (Pietiäinen et al. 2024), where we have tested fiber addition levels 0-5%. In the study, lower fiber addition levels did not show as clear differences in bread quality. Therefore, 5% was chosen for this study to be able to see differences between different fiber fractions and understand the relationship between fraction properties and dough quality better. We agree with the reviewer that this connection to previously published work was not expressed clear enough and we have clarified this both in the introduction (lines 52-56) and methods (lines 187-189) of this manuscript.

Comment 2: Hydrolysis time: 24 h, how about 2-36 h?

Response 2: The hydrolysis procedure used in this study has previously been optimized by Ruthes et al. 2017, where the authors studied hydrolysis of AX fractions. Therefore, we have decided to not focus on optimization of the hydrolysis process in this study. We agree with the reviewer that the link to previously done optimization was not clearly indicated in the text and we have clarified the connection to the previous study in the text (lines 101-103).

Comment 3:  There is no significantly different between the microstructures of wheat dough.

Response 3: We agree with the reviewer that based on the presented images we cannot state that there is a significant difference between the different doughs as no image analysis was done on the microscopy results. We have reformulated the tone of the discussion both in results (line 402) and conclusions (lines 441-442) to be clearer that these are only indications rather than statistically significant results.

Comment 4: If AX can be used in wheat dough, what are the usage limits?

Response 4: We interpret the reviewer to refer to the upper limit of incorporated AX in wheat dough before the fiber addition starts to have detrimental effects on dough quality. Based on our previous work (Pietiäinen et al 2022, Pietiäinen et al. 2024), AX addition starts to show a negative effect on dough and bread quality indicators with AX addition levels of 5% or more. That is one of the reasons why we have chosen to use the addition level of 5% in this project. We agree with the reviewer that this was not clearly expressed in the text, and we have clarified both the addition level and effect of fiber in the introduction (lines 48-50 and 52-56).

Comment 5: The results cannot support the conclusions.

Response 5: We agree with the reviewer that the tone of the conclusions was overly positive on comparison to the results. Therefore, we have changed the conclusions to be more specific with wording in terms of what we mean by dough quality (lines 425-426) and rephrased some sentences to indicate more doubt about the results (lines 441-442, 446-447). We have also added a section on the limitations of this study in conclusions (lines 447-449).

Reviewer 2 Report

Comments and Suggestions for Authors

The authors studied the effects of acylation and hydrolysis of wheat bran Ferulate on dough quality and microstructure. This study is meaningful, but there are many problems that need to be modified.

1. The repetition rate of this paper is high, and the author must revise the paper. The repetition rate of a single paper has reached 14%, and it must be reduced to less than 10%.

2. None of the keywords come from the title. The composition of keywords should be a combination of part from the title and part from the text content.

3. Section 2.1. What is the purity of the other materials? Analytic reagent? Such as NaOH, ethanol.

4. The data in the table should be expressed as mean ± standard deviation. For example, 75.9 (±6.6) should be 75.9 ± 6.6.

5. The pictures in the article do not meet the requirements of the journal, such as Figure 1, why the ordinate only has numbers? Authors should take a good look at the articles that have been published in the journal, and then make changes to the pictures and tables.

6. Figure 3 is too ugly.

7. In the conclusion, the author should mention the unresolved issues of this study and the research directions that need to be focused in the future.

Author Response

Reviewer 2

The authors studied the effects of acylation and hydrolysis of wheat bran Ferulate on dough quality and microstructure. This study is meaningful, but there are many problems that need to be modified.

We would like to thank reviewer #2 for their thoughtful review. We appreciate the detailed comments regarding the content and formatting of the manuscript that have helped to improve the quality of this work.

Comment 1: The repetition rate of this paper is high, and the author must revise the paper. The repetition rate of a single paper has reached 14%, and it must be reduced to less than 10%.

Response 1: We agree with the reviewer that this manuscript contained too much repetition. We believe this to be largely due to the methods section, as we have used same methods in our previous studies. We have therefore rephrased large parts of the methods section (lines 94-99, 101-105, 122-129, 165, 184-194, 231-237).

Comment 2: None of the keywords come from the title. The composition of keywords should be a combination of part from the title and part from the text content.

Response 2: We agree with the reviewer and we have changed all the keywords to be from the abstract (line 32).

Comment 3: Section 2.1. What is the purity of the other materials? Analytic reagent? Such as NaOH, ethanol.

Response 3: We agree with the reviewer, and we have added more detailed descriptions of materials throughout the section on materials and methods (lines 90-91, 96, 97, 99, 104, 137, 141-142, 144, 225). The purity and chemical composition of the original AX fraction FAX was measured in this study with the other fiber fractions and details can be found in table 1.

Comment 4: The data in the table should be expressed as mean ± standard deviation. For example, 75.9 (±6.6) should be 75.9 ± 6.6.

Response 4: We agree with the reviewer and all data in tables is now expressed mean ± standard deviation in all tables (Table 1, 2, 3).

Comment 5: The pictures in the article do not meet the requirements of the journal, such as Figure 1, why the ordinate only has numbers? Authors should take a good look at the articles that have been published in the journal, and then make changes to the pictures and tables.

Response 5: We agree that the resolution and formatting of the figures in this manuscript was not uniform and had room for improvement. We have added information about statistical differences, harmonized the formatting and colors in all figures, and increased the resolution (Figure 1-4).  The ordinate of the figure 1 did already include an axis title with the clarification of measured compound and unit of measure but we hope it is more clearly visible now with new formatting.

Comment 6: Figure 3 is too ugly.

Response 6: We have improved the visual appearance of figure 3 by increasing the clarity and harmonizing the colors and formatting. We have also added statistical differences to the image to make it more clear to the reader. We believe that the resolution of this image was too low, and we hope the higher resolution will improve the appearance.

Comment 7: In the conclusion, the author should mention the unresolved issues of this study and the research directions that need to be focused in the future.

Response 7: We have added a section to the conclusions including issues related to this study and future research directions (lines 447-449).

Reviewer 3 Report

Comments and Suggestions for Authors

This manuscript is on the study of the application of arabinoxylan extracted from wheat bran in food products (breads), It is an interesting work. I enjoyed reading it. It should be publishable in Foods after minor revision.

Specific comments:

Title and the whole manuscript. You may need to be more specific what specific quality you are referring to when you use the term “dough quality”. For example, Line 18, you claimed “… to affect dough quality”.  but later, you have a conflicting conclusion “doest not appear to affect dough quality” (Line29).  Indeed, not affect all dough quality parameters? Or some of them?

Methods. 2.3.3 to 2.3.6. These subsections are too short. If they are too simple, consider to combine them into one subsection (e.g. basic nutrient parameters or alike). Also double check the accuracy of your description. Such as,  the protein content section is not accurate described. If should be “The protein content of fractions was determined by the total N content with …”

2.5. “Dough properties” is same to “Dough quality” or not? Again should be more specofic. Physical properties, Chemical, properties, or physicochemical properties?  

Table 1. Clarify if these values are based on wet or dry basis. Also for other Tables/figures.

Author Response

Reviewer 3

This manuscript is on the study of the application of arabinoxylan extracted from wheat bran in food products (breads), It is an interesting work. I enjoyed reading it. It should be publishable in Foods after minor revision.

We would like to thank reviewer #3 for their thoughtful review. We appreciate the comments regarding the content of the manuscript that have helped to improve the quality of this work.

Comment 1: Title and the whole manuscript. You may need to be more specific what specific quality you are referring to when you use the term “dough quality”. For example, Line 18, you claimed “… to affect dough quality”.  but later, you have a conflicting conclusion “doest not appear to affect dough quality” (Line29).  Indeed, not affect all dough quality parameters? Or some of them?

Response 1: We agree with the reviewer that the term “dough quality” was not specific enough. We have changed the wording from simply reading dough quality throughout the text, including the title, to refer to dough rheology as this work mainly focuses on dough rheology (lines 3, 18, 29-30, 69, 75, 423-424). AX can have either negative or positive effect on dough rheology depending on AX properties and addition level, and we have clarified this in the introduction (lines 48-59).

Comment 2: Methods. 2.3.3 to 2.3.6. These subsections are too short. If they are too simple, consider to combine them into one subsection (e.g. basic nutrient parameters or alike). Also double check the accuracy of your description. Such as,  the protein content section is not accurate described. If should be “The protein content of fractions was determined by the total N content with …”

Response 2: We agree with the reviewer, and we have combined methods for protein, ash, starch and B-glucan analysis into one section (lines 121-129). We have also increased the accuracy based on the comment and rephrased the line on protein content determination according to reviewer’s suggestion (line 123-124).

Comment 3: 2.5. “Dough properties” is same to “Dough quality” or not? Again should be more specofic. Physical properties, Chemical, properties, or physicochemical properties? 

Response 3: We agree with the reviewer that the terminology was inconsistent, and we have renamed this section to “Dough quality”. As this is only a general title for the section including all methods used for measuring dough properties, we believe that the subtitles for the subsections of this section clarify which parameters we are referring to with “dough quality”.  We have also clarified the term dough quality in text to refer to the dough properties in question throughout the text (lines 3, 18, 29-30, 69, 75, 423-424).

Comment 4: Table 1. Clarify if these values are based on wet or dry basis. Also for other Tables/figures.

Response 4: We agree with the reviewer and have added the mention that all values are dry weight basis to all tables where applicable (line 259 and 283).

Reviewer 4 Report

Comments and Suggestions for Authors

Feruloylation and hydrolysis of arabinoxylan extracted from wheat bran: Effect on dough quality and microstructure.

Line 42: the authors can mention what are the negative effect on bread when AX is added.

Line 51: what properties can AX improve in bread when is used as ingredient

Line 91: Acetic acid was used; however, the authors did not mention it concentration

Line 93: I suggest to deleted the text pore size of in ( )

Line 97: the pH of the AX and FAX solution was adjusted with what acid and concentration to reach pH 5.0

Line 98, 106 and 138: the incubation of mixes was at what temperature, the authors are omitting the data

Line 105: the correct formula of sulfuric acid is H2SO4

Line 112: there is a space in 4 × 250

Line 140: the authors are omitting concentration and volume of NaCl

Line 142 and 162: temperature time and rpm of centrifuged are missing

Line 151-152: there are spaces in 2.1 x 150

Table 1,2,3: standard deviation I suggest to express it without ( )

Line 233: I suggest to change the text “were similar 233 to each other”, because the authors did a statistics analysis.

3.1. Fraction composition: authors mentioned before that they did a statistics analysis, so they can describe if there is a significant difference among treatment, I suggest to express the results in this section having in mind the statistics analysis.

Figure 1: the color of the letters is grey no black

Line 298: there is a space 63.3 %,

Line 414: the authors mentioned “affect the quality”, how can the authors describe this effect with the results that they show?

I suggest to look for more papers and improve the discussion of the results.

Author Response

Reviewer 4

We would like to thank reviewer #4 for their thoughtful review. We appreciate the detailed comments regarding the formatting and content of the manuscript that have helped to improve the quality of this work.

Comment 1: Line 42: the authors can mention what are the negative effect on bread when AX is added.

Response 1: We have added a mention of the negative effects of AX addition to dough to introduction (48-50).

Comment 2: Line 51: what properties can AX improve in bread when is used as ingredient

Response 2: In our previous study using hydrolyzed and feruloylated AX fraction, we have observed them to improve volume and crumb structure compared to unhydrolyzed and unferuloylated AX fractions (Pietiäinen et al. 2024). We have added a mention of this to the introduction (lines 52-56).

Comment 3: Line 91: Acetic acid was used; however, the authors did not mention it concentration

Response 3: We agree with the reviewer that details were missing from the materials and methods regarding concentrations. We have added details on acetic acid concentrations on the suggested place (line 97) but also throughout materials and methods section (lines 99, 104, 144).

Comment 4: Line 93: I suggest to deleted the text pore size of in ( )

Response 4: We have removed text “pore size” from brackets (line 99).

Comment 5: Line 97: the pH of the AX and FAX solution was adjusted with what acid and concentration to reach pH 5.0

Response 5: We agree with the reviewer that details were missing from the materials and methods section. We have added details on the acid used (line 104) and on other materials used throughout the section (lines 97, 99, 144).

Comment 6: Line 98, 106 and 138: the incubation of mixes was at what temperature, the authors are omitting the data

Response 6: We agree with the reviewer that details were missing from the materials and methods section. We have added details on incubation times and temperatures used throughout the section (lines 96, 98, 104, 112, 142, 171).

Comment 7: Line 105: the correct formula of sulfuric acid is H2SO4

Response 7: We have corrected the formula in the text (line 111).

Comment 8: Line 112: there is a space in 4 × 250

Response 8: We have removed the spaces (line 118).

Comment 9: Line 140: the authors are omitting concentration and volume of NaCl

Response 9: We have added the concentration and amount of NaCl (line 144).

Comment 10: Line 142 and 162: temperature time and rpm of centrifuged are missing

Response 10: We have added temperature and rpm for centrifugations (lines 146-147, 171)

Comment 11: Line 151-152: there are spaces in 2.1 x 150

Response 11: We have removed the spaces (line 156).

Comment 12: Table 1,2,3: standard deviation I suggest to express it without ( )

Response 12: We agree with the reviewer and have removed the brackets from all tables (table 1-3).

Comment 13: Line 233: I suggest to change the text “were similar 233 to each other”, because the authors did a statistics analysis.

Response 13: We agree with the reviewer and we have rephrased the sentence and clarified the statistical significance of the differences (line 241-242).

Comment 14: 3.1. Fraction composition: authors mentioned before that they did a statistics analysis, so they can describe if there is a significant difference among treatment, I suggest to express the results in this section having in mind the statistics analysis.

Response 14: We agree with the reviewer that all the necessary information about statistical differences was not previously available in the manuscript. We have added details about statistics in all the tables and figures where applicable (Tables 1-3, Figure 1).

Comment 15: Figure 1: the color of the letters is grey no black

Response 15: We have changed the color to black (Figure 1).

Comment 16: Line 298: there is a space 63.3 %,

Response 16: We have removed the space (line 305).

Comment 17: Line 414: the authors mentioned “affect the quality”, how can the authors describe this effect with the results that they show?

Response 17: We agree that the term “dough quality” has not been specific enough and changed the wording in conclusions (423). We have also clarified the specific dough quality in question throughout the text (lines 3, 18, 29-30, 69, 75)

Comment 18: I suggest to look for more papers and improve the discussion of the results.

Response 18: We have added two new references (lines 562-565) and discussion on the effect of added water on dough microstructure (lines 413-416), as the authors felt that to be the section needed most clarification based on reviewers comments.

Round 2

Reviewer 1 Report

Comments and Suggestions for Authors

The manuscript has undergone some revisions, and there are still some writing issues that need to be adjusted, such as inconsistent units of centrifugal speed, g, rpm. Some values do not have intervals between units, and all graph axes are not clearly displayed.

Author Response

Comment 1: The manuscript has undergone some revisions, and there are still some writing issues that need to be adjusted, such as inconsistent units of centrifugal speed, g, rpm. Some values do not have intervals between units, and all graph axes are not clearly displayed.

Response 1: We would like to thank reviewer #1 for their thoughtful review, and we appreciate the detailed comments regarding the formatting of the manuscript. We have now carefully revised the manuscript and added a space between all values and units throughout the manuscript. We have changed the rpm to g (line 171) and harmonized unit s-1 to Hz (line 181). We have also revised figures 1-3 by adding graph axes to these figures.

Reviewer 2 Report

Comments and Suggestions for Authors

Figure 1 drawing is not standardized, horizontal and vertical coordinate lines are missing.

Author Response

Comment 1: Figure 1 drawing is not standardized, horizontal and vertical coordinate lines are missing.

Response 1: We would like to thank reviewer #2 for their thoughtful review, and we appreciate the detailed comments regarding the formatting of the figures. We have revised figures 1-3 and added graph axes to these figures.